# Antidiabetic Properties of Chitosan and Its Derivatives

**DOI:** 10.3390/md20120784

**Published:** 2022-12-17

**Authors:** Huei-Ping Tzeng, Shing-Hwa Liu, Meng-Tsan Chiang

**Affiliations:** 1Institute of Toxicology, College of Medicine, National Taiwan University, Taipei 10051, Taiwan; 2Department of Pediatrics, College of Medicine and Hospital, National Taiwan University, Taipei 10051, Taiwan; 3Department of Medical Research, China Medical University Hospital, China Medical University, Taichung 40402, Taiwan; 4Department of Food Science, National Taiwan Ocean University, Keelung 20224, Taiwan

**Keywords:** chitosan, chitosan oligosaccharide, diabetes mellitus

## Abstract

Diabetes mellitus is a chronic metabolic disorder. In addition to taking medication, adjusting the composition of the diet is also considered one of the effective methods to control the levels of blood glucose. Chitosan and its derivatives are natural and versatile biomaterials with health benefits. Chitosan has the potential to alleviate diabetic hyperglycemia by reducing hepatic gluconeogenesis and increasing skeletal muscle glucose uptake and utility. Scientists also focus on the glucose-lowering effect of chitosan oligosaccharide (COS). COS supplementation has the potential to alleviate abnormal glucose metabolism in diabetic rats by inhibiting gluconeogenesis and lipid peroxidation in the liver. Both high and low molecular weight chitosan feeding reduced insulin resistance by inhibiting lipid accumulation in the liver and adipose tissue and ameliorating chronic inflammation in diabetic rats. COS can reduce insulin resistance but has less ability to reduce hepatic lipids in diabetic rats. A clinical trial showed that a 3-month administration of chitosan increased insulin sensitivity and decreased body weight and triglycerides in obese patients. Chitosan and COS are considered Generally Recognized as Safe; however, they are still considered to be of safety concerns. This review highlights recent advances of chitosan and its derivatives in the glucose-lowering/antidiabetic effects and the safety.

## 1. Introduction

### 1.1. Diabetes Mellitus (DM)

DM is a chronic metabolic disorder with an increasing prevalence since last century, affecting 537 million adults (1 in 10) worldwide in 2021. The International Diabetes Federation (IDF) estimated that this number will rise to 643 million by 2030 and 783 million by 2045 [1]. According to WHO’s 2019 global health estimates, DM has entered the top 10 causes of death, with a significant increase of 70% globally over the period from 2000 to 2009 [2]. This highlights the truth that DM has become an accelerating public health concern and a substantial burden on society [3]. In this regard, diagnosing prediabetes or diabetes early and accurately is, therefore, pivotal for physicians to trigger the proper treatment and control the further progress. Fasting plasma glucose, a 2 h oral glucose tolerance test (OGTT) and glycated hemoglobin A1c (HbA1c) are the standard measures for the diagnosis of type 2 DM (T2DM). More recently, some nonclassical markers of hyperglycemia, including fructosamine, glycated albumin, and 1,5-anhydroglucitol, have been used as adjuncts to standard measures to obtain a more detailed understanding of the alterations of glycemic control [4]. DM is characterized by hyperglycemia, attributed to unbalanced glucose homeostasis by immune-mediated insulin deficiency (type 1 DM, T1DM), or by insulin resistance and disturbed insulin secretion (T2DM) [5]. To maintain normal blood glucose levels, insulin secreted by pancreatic β-cells normally suppresses glucose output from the liver, promote glucose uptake by skeletal muscle and adipose tissue, and reduce lipolysis and fatty acid release from adipose tissue [6,7]. Decreased insulin secretion or resistance to insulin action will, therefore, affect insulin signaling in those target tissues, which may lead to impaired glucose tolerance and hyperglycemia of diabetes once β-cells are unable to compensate fully for the decreased insulin sensitivity [8]. Therefore, the key strategy for DM management is to maintain blood glucose levels in the target range, as uncontrolled hyperglycemia may lead to chronic complications, such as cardiovascular disease, nephropathy, retinopathy, and neuropathy [9]. Subcutaneous insulin administration is needed to control blood glucose levels in T1DM and in advanced T2DM. However, frequent subcutaneous injections can contribute to poor patient adherence, and tight glycemic control remains a constant challenge for those with DM [10]. Therefore, complementary and alternative therapies for DM management have been researched for years. A variety of bioactive products derived from plants, animals, and fungi have been shown to possess anti-diabetic or hypoglycemic potential, such as plant-based curcumin, resveratrol, *Teucrium polium* aerial parts, and *Artemisia campestris* extract, as well as animal-based fish oil, milk casein, and chitosan [11,12,13,14,15]. Obesity, known for disordered lipid metabolism and insulin resistance, is recognized as one of the major risk factors for the development of T2DM, which accounts for 90–95% of all diagnosed cases of diabetes [16,17]. To date, in addition to pharmacological treatment, lifestyle modifications, such as weight loss, regular physical activity and healthy eating with a calorie restricted diet or high fiber intake, are believed to be beneficial in the prevention and management of T2DM and its possible related complications [7,18]. 

### 1.2. Chitosan

Chitin is a cellulose-like polysaccharide, comprised of repeating lineal β-(1,4) N-acetyl-d-glucosamine (GlcNAc) monomers [19]. It is the second most abundant natural biopolymer in the world after cellulose and can be found in exoskeletons of crustaceans and in molluscan organs, fungi, insects, and yeasts [19]. Chitosan, a partially degraded product from N-deacetylation of chitin under alkaline condition, is a random copolymer of N-acetyl-D-glucosamine and d-glucosamine units [20]. The proportion of the d-glucosamine units defines the degree of deacetylation (DD). The DD reaching around 50% demonstrates the generation of chitosan, as it begins to dissolve at acidic pH; otherwise, it remains as chitin [21]. The free amino groups, which make chitosan a cationic polymer, can be used for chemical modifications and conjugations with other biopolymers into multiple chitosan derivatives, as well as for allowing it to be dissolved in acidic conditions with amino groups protonated [21]. While in acidic pH, such as in the stomach, the positive charge on chitosan can interact with the negative charges on fatty acids, bile acids, protein, lipids, phospholipids and intracellular molecules with negative charges, such as deoxyribonucleic acid (DNA) [22,23]. Thus, chitosan and its modified derivatives are bioactive polymers with wide ranges of applications in biomedicine, agriculture, food processing, and non-food chemical industries, and the global chitosan market size was valued at USD 6.8 billion in 2019 [19,24]. Due to their unique biological characteristics, which are biocompatible, biodegradable, and generally nontoxic in the body, and combined with their various biological activities (antimicrobial, antioxidant, and anti-inflammatory), they are intensively used in biomedical and pharmaceutical fields, such as drug delivery, artificial skin, wound dressing, and biofilms [25,26]. Recently, rather than serving as materials in these biomedical applications, chitosan and its derivatives have attracted rising interests for their health benefits in the development and management of diabetes [27,28]. 

Varying types and different molecular weights (MW) of chitosan have been used to study their health impact in humans and in animals. Numerous studies have shown that their biological properties are largely corelated to their MW, DD, degree of polymerization (DP), and charge distribution [29]. To date, there is no distinct standard definition of chitosan. Rinaudo et al. suggested that chitin with DD values reaching 50% are called chitosan [21]. However, Kou et al. reported that the DD of chitosan was between 40% and 75% from different sources of starting materials, and most commercially available chitosan has DD between 70% and 90%, as higher DD increases the number of positive charges and shows greater biological properties [30,31]. In addition, DP or MW are other important factors associated with its bioactivities [26]. It is generally believed that lower MW chitosan may have stronger bioactivity than higher MW chitosan. In terms of antimicrobial activity against *Escherichia coli*, chitosan with a low MW between 10 and 25 kDa has a greater effect than high MW chitosan (MW > 250 kDa) [32]. The anti-obesity effect of MW 46 kDa chitosan is better than that of 130 kDa and 21 kDa chitosan [33]. However, a study has reported that high MW chitosan (740 kDa) exhibits a higher inhibitory effect on intestinal lipid absorption than low MW chitosan (91 kDa), indicating the regulation of the biological effects of chitosan is, no doubt, complex [34].

### 1.3. Characterization of Chitosan Oligosaccharide (COS)

While chitosan has a DD higher than 90%, a DP less than 20, and an average MW less than 3.9 kDa, it is usually referred as chitosan oligomer, or COS, which is a hydrolyzed product of chitosan or chitin by a method of enzymatic or chemical hydrolysis involving deacetylation and depolymerization [29,35]. The same as chitosan, there is no clear-cut definition of COS. Muanprasat and Chatsudthipong suggested that COS had a DP < 50–55 and an average MW < 10 kDa [36]. Compared with chitosan, COS possesses much greater solubility and lower viscosity under physiological conditions, which is attributed to its shorter chain lengths and higher DD exposing more free amino groups [37]. Moreover, it permeates through intestinal epithelial barriers freely and is absorbed into the circulation [26]. Therefore, COS has recently become a promising therapeutic agent of natural origin, with high solubility and absorbability properties, as well as versatile biological functions. Interesting, COS with DP > 6 has been testified to possess greater bioactivities, including antimicrobial, anti-tumor, and immunopotentiation, as compared to a smaller COS [38]. Overall, apart from having characteristics of chitosan, COS is generally believed to be biologically more active than chitosan.

Over the past few decades, several emerging studies have been conducted to evaluate the benefits of chitosan and its derivatives in diabetic control; however, the blood glucose-lowering effects and the underling mechanisms are still in progress. Diverse mechanisms might be involved in the anti-diabetic effects of chitosan. As such, this review aims to discuss the latest research progress on their potential effects against diabetes mainly from in vivo rodent studies. Moreover, the possible mechanisms underlying these benefits of chitosan are addressed in particular. We also include some updated safety evaluation of chitosan and its derivatives. The information and data were collected by a systematic survey of the literature using PubMed, Web of Science, and Google Scholar, as well as by our research group. We hope that this review will help the readers learn about the potential of chitosan and its derivatives for improving glycemic control and maintaining glucose homeostasis, which may be helpful for future research in related fields, ultimately benefitting the health of diabetic patients.

## 2. Anti-Diabetic Effect and Mechanism of Chitosan and Its Derivatives

The anti-diabetic activity of chitosan and its derivatives has been investigated by different types of diabetic models. Evidence suggests that the effects of chitosan and its derivatives on improving the pathophysiology of diabetes are involved in multiple pathways. In this section, we mainly focused on relevant studies, and summarized the antidiabetic activities of chitosan and its derivatives, including glucose lowering, lipid lowering, anti-oxidative, anti-inflammatory and gut regulating effects.

### 2.1. Regulation of Glucose Homeostasis

Hyperglycemia observed in various diabetic models can be reduced by chitosan and its derivatives. Chitosan was observed to have hypoglycemic effects in neonatal streptozotocin (STZ)-induced diabetic mice [39]. COS with a MW of 20 kDa markedly delayed the rise of blood glucose levels in low-dose STZ-induced slowly progressive diabetic mice, and in genetically obese diabetic KK-Ay mice as well [40,41]. After 8 weeks of oral gavage of COS with a MW of 1.5 kDa, T2DM rats induced by a high-energy diet and a low-dose STZ injection exhibited a reduction in fasting blood glucose levels and improved glucose tolerance [42]. Long-term (42 days) administration of COS with a MW of < 1 kDa reduced the fasting blood glucose and HbA1c levels in obese diabetic *db/db* mice [43]. In high-fat-diet-fed mice, N-acetylated COS (97% DD) improved the metabolic syndrome, including lowering blood glucose [44]. Moreover, COS has a hypoglycemic effect in alloxan-induced diabetic mice [27]. As glucose homeostasis is regulated by various pathways and is cooperated by different organs and tissues, the participating molecules and signalings underlying the anti-hyperglycemic action of chitosan are listed.

#### 2.1.1. Chitosan Protects Pancreatic β-Cells and Promotes Insulin Secretion

COS with a molecular mass of 1200 u has been found to accelerate the proliferation of β-cells in vitro, and to improve the impaired pancreatic function in STZ-induced diabetic rats, facilitating the insulin release and glycemic control [45]. It was further shown that COS could prohibit the apoptosis of pancreatic islet cells in vitro, as well as improve the antioxidant ability and protect the pancreas against oxidative damage caused by STZ [46]. Furthermore, Ju et al. reported that COS has antidiabetic properties in pancreatic INS-1 β-cells by enhancing proliferation, triggering glucose-stimulated insulin release, upregulating glucose transporter 2 (GLUT2) mRNA expression, and preventing STZ-induced apoptosis [42]. 

#### 2.1.2. Chitosan Alleviates Insulin Resistance and Leptin Resistance 

Insulin resistance is one of the contributing factors to the development of T2DM, and a hallmark of T2DM as well. Numerous animal studies have proven that chitosan and its derivatives can improve insulin resistance and ameliorate glucose intolerance [47,48,49]. COS treatment can effectively alleviate the general health, relieve symptoms, normalize blood glucose levels, and improve insulin sensitivity in diabetic rats [42]. 

Leptin resistance, which is characterized by elevated circulating levels of leptin and reduced leptin sensitivity, is not only a phenomenon of obesity, but also contributes to the development and maintenance of obesity [50]. Moreover, leptin has been demonstrated to have effects on hyperglycemia control, mainly through its action on the central nervous system. However, a recent study has reported that hepatic leptin signaling suppressed gluconeogenesis to maintain glucose homeostasis [51]. Supplementation of high-MW chitosan (380 kDa) has been found to cause a decrease in plasma leptin levels in a high fructose-induced glucose intolerant rat model [48]. Pan et al. found that hepatic leptin resistance in diet-induced obese rats could be attenuated by COS (MW ≤ 1 kDa) administration via upregulating hepatic leptin receptor-b (LepRb) expression and activating its downstream JAK2-STAT3 signaling, which may then facilitate leptin signaling to suppress gluconeogenesis [52].

#### 2.1.3. Chitosan Promotes Glucose Uptake and Storage in Peripheral Tissues

The failure of peripheral tissues of the body to properly uptake and utilize glucose may result in chronic hyperglycemia. The skeletal muscle and adipose tissue are the two major glucose uptake targets sensitive to insulin regulation. Glucose transporter 4 (GLUT4), which is mainly sequestered intracellularly and translocated to the cell membranes in the presence of insulin or other stimuli, plays a pivotal role in regulating glucose uptake into muscle and adipose, and maintaining normal glucose homeostasis. Thus, defects of GLUT4 may lead to insulin resistance and glucose intolerance [53,54]. The GLUT4 gene expression in soleus muscle and adipose tissue were upregulated by COS treatment in T2DM rats [36]. Moreover, both low- and high-MW chitosan enhance muscle glucose uptake by promoting GLUT4 translocation from cytoplasm to membranes in STZ-induced T1DM rats, which might be mediated by increased AKT phosphorylation [55]. Additionally, COS significantly increases glycogen synthesis in the liver by enhancing glucokinase activity, indicating that COS facilitates more blood glucose to be stored in the liver as glycogen [42]. Liu et al. also observed that both low- and high-MW chitosan were capable of increasing glycogen content in the STZ-induced diabetic liver [55].

#### 2.1.4. Chitosan Inhibits Carbohydrate-Hydrolyzing Enzymes and Hepatic Gluconeogenesis

The inhibition of enzymes responsible for carbohydrate digestion would lead to their slower absorption in the diet, which would then reduce the magnitude of postprandial glucose and insulin responses to dietary carbohydrates, which has been proved to be clinically useful in the management of diabetic individuals [56]. An in vitro study has found that COS with MW < 1 kDa (GO2KA1) may prevent hyperglycemia by inhibiting intestinal glucose digestion-related enzyme α-glucosidase and the transporters sodium-dependent glucose cotransporter (SGLT)-1 and GLUT2, and enhance glucose uptake, at least in part, by upregulating adiponectin expression through peroxisome proliferator-activated receptor (PPAR)-γ in adipocytes [57]. A study has reported that in STZ-induced diabetic rats, high-MW chitosan had broader inhibitory effects on intestinal disaccharidase, including sucrase, lactase and maltase, whereas low- MW chitosan only decreased lactase activity [58]. Chiu et al. showed that low-MW chitosan (80 kDa) mitigated the increased intestinal disaccharidase or α-glucosidases activity in high-fat-diet-induced obese rats, but COS with MW 719 Da supplementation had no such inhibitory effects [59]. Altogether, as a carbohydrate blocker, chitosan is capable of reducing plasma glucose by delaying the absorption of glucose and then improving the insulin sensitivity of peripheral tissues.

Obesity-related hyperglycemia can be largely attributed to the increased hepatic gluconeogenesis, which is mainly controlled by insulin signaling [60]. Both low- and high-MW chitosan can reduce liver gluconeogenesis, as evidenced by inhibiting the expression of gluconeogenesis-related enzyme, phosphoenolpyruvate carboxykinase (PEPCK), thus ameliorating abnormal glucose and insulin profiles in STZ-induced T1DM rats [55]. They also found that chitosan increased phosphorylation of adenosine monophosphate (AMP)-activated protein kinase (AMPK) in the liver of diabetic rats, which acts as a cellular energy sensor. Once activated by energy stress, AMPK restores cellular energy balance by promoting catabolic pathways to generate ATP, as well as suppressing anabolic pathways, such as downregulating the transcription of gluconeogenic enzymes [61]. Taken together, the inhibition of gluconeogenesis by chitosan might be, at least in part, through regulating AMPK activation.

### 2.2. Regulation of Dyslipidemia and Metabolic Disorders Associated with Diabetes

As obesity is a major risk factor for the development T2DM, the potential of chitosan against hyperlipidemia and related metabolic disorders may help to delay and manage the development and complications of T2DM [15]. A meta-analysis has shown that chitosan can slightly reduce body weight and improve blood lipid profile and blood pressure in overweight and obese patients [62]. Many molecules and signaling pathways contribute to the benefits of chitosan on lipid metabolism and we have highlighted certain mechanisms that we are interested in. 

#### 2.2.1. Chitosan Regulates Lipid Absorption and Excretion

Chitosan and its derivatives have been known for their lipid-lowering effects. A meta-analysis from murine models elucidated that chitosan was significantly effective in lowering total cholesterol (TC) and triglyceride (TG) in the blood and liver and increasing fecal excretion of TC and TG, which demonstrates that chitosan is beneficial in improving the disturbance of the lipid metabolism [63]. A study has shown that high-MW (1000 kDa) chitosan exerts better antidiabetic activity in STZ-induced diabetic rats, as evidenced by a decrease in blood glucose levels, a decrease in plasma and liver TC and TG levels, and an increase in plasma HDL levels, whereas low-MW (140 kDa) chitosan has less or no effects on those parameters [58]. A study further showed that high-MW chitosan exhibited a higher efficiency than low-MW chitosan on the inhibition of intestinal lipid absorption and the stimulation of hepatic fatty acid oxidation, which may lead to an improvement in hepatic lipid accumulation [34]. Moreover, supplementation with either low- or high-MW chitosan could counteract the increased microsomal triglyceride transfer protein (MTTP) protein expression and the decreased angiopoietin-like protein-4 (Angptl4) expression in the intestines of rats fed with high-fat diets, both of which play critical roles in intestinal lipid absorption and digestion. These results reveal that chitosan can mediate lipid homeostasis by regulating both the absorption and excretion of lipids. 

#### 2.2.2. Chitosan Alleviates Hepatic Steatosis

Nonalcoholic fatty liver disease (NAFLD), known as hepatic steatosis, is highly associated with metabolic disorder that developed from insulin resistance-induced hepatic lipogenesis [64]. In STZ/nicotinamide-induced T2DM rats, hepatic lipids, including TG and TC content, and HOMA-IR value were found to be reduced by high-MW chitosan (830 kDa) treatment [47]. In addition to the decrease in hepatic lipids, the elevation of fecal lipid excretion by high-MW chitosan was observed as well [48]. This effect can be explained by the ability of high-MW chitosan (560 kDa) to suppress the expression of lipogenesis-associated genes, including fatty acid synthase (FAS) and 3-hydroxy-3-methylglutaryl coenzyme A reductase (HMGCR), by activation of AMPK and inhibition of its downstream lipogenic transcription factors (i.e., PPARγ) and sterol regulatory element-binding protein 1c (SREBP1c), and, in turn, attenuate TG and TC accumulation in the liver, and finally increase fat excretion in the feces [65]. 

Hepatic PPARα and its downstream molecules, such as acyl-CoA oxidase (AOX1), 3-hydroxy-3-methylglutaryl coenzyme A reductase (HMGCR), and cytochrome P450-7A1 (CYP7A1), are also key mediators in the regulation of lipid homeostasis. In a high-fat-diet-induced obese rat model, high-MW chitosan (642 kDa) has been shown to upregulate the protein expression of PPARα and the gene expression of AOX1, which are involved in the β oxidation of fatty acids [66]. Moreover, HMGCR and CYP7A1, which are involved in cholesterol synthesis and the transformation of cholesterol into bile acid, are downregulated by chitosan supplementation [67]. These findings suggested that chitosan could facilitate hepatic β oxidation capacity and alter hepatic abnormal cholesterol metabolism and improve the progression of NAFLD.

It is worth noting that adropin, a peptide hormone secreted primarily by the brain and liver, is involved in glucose and lipid homeostasis [68]. It has been shown to be capable of protecting against NFALD and hyperinsulinemia in adropin-overexpressing transgenic mice fed with high-fat diet [69]. Gao et al. showed that adropin improved glucose tolerance and enhanced insulin sensitivity in diet-induced obese mice, along with the preferential utilization of carbohydrate for energy production in muscles [70]. Moreover, they recently reported that hepatic glucose metabolism and hepatic insulin sensitivity were controlled by adropin [71]. Moreover, in a rat model of T2DM, adropin was found to reduce blood glucose levels, improve insulin sensitivity, ameliorate lipid profile, and suppress inflammatory marker (TNF-α, IL-6, and iNOS) expression [72]. Thus, it would be interesting to see whether adropin plays a role in chitosan-mediated anti-diabetic effects.

#### 2.2.3. Chitosan Attenuates Obese Adipose Tissue 

In addition to lowering hepatic lipid accumulation, high-MW chitosan (830 kDa) administration reduced lipid accumulation in adipose tissue, as evidenced by a suppression of adipocyte hypertrophy, TG accumulation, and lipoprotein lipase (LPL) activity, as well as an increase in the lipolysis rate, in STZ/nicotinamide-induced diabetic rats [47]. The gene expression of the fatty acid transport-related molecules, fatty acid transport proteins (FATPs), and fatty acid-binding proteins (FABPs) were also inhibited by chitosan. The mechanism underlying the inhibition of FATP and FABPs by chitosan might be through AMPK activation, as well as PPARγ and SREBP1c inhibition [65]. It is well known that LPL, which is activated by insulin, removes fatty acids from chylomicron and VLDL into adipocytes through FABP and FATP transportation [73]. Therefore, it can be postulated that chitosan attenuates adipocyte hypertrophy and lipid storage by suppressing the process of fatty acid uptake via inhibiting the LPL activity and FABP and FATP expression. 

Likewise, PPARγ, cooperating with CCAAT-enhancer binding protein α (C/EBPα), is required for preadipocyte to differentiate into mature adipocytes; therefore, PPARγ is considered a master regulator of the adipogenesis [74]. As evidenced by fewer and smaller adipocytes in response to COS supplementation, adipogenesis could be reduced by COS in high-fat-diet-fed rats, through downregulating the gene expression of PPARγ and C/EBPα [52]. Moreover, it is known that high-fat/cholesterol-diet-induced liver X receptor (LXR) activation may upregulate SREBP1c, which, in turn, stimulates fatty acid synthase (FAS) to synthesize fatty acids, resulting in TG production, as well as an upregulation of ATP-binding cassette subfamily A member-1 (ABCA1), which catalyzes the efflux of excess cholesterol [75]. Chiu et al. also observed the inhibition of LXRα, SREBP1c, and FAS expression and the increase in ABCA1 expression by high-MW chitosan, compared to the high-fat diet group [66]. Collectively, these results suggest that the inhibitory effects of chitosan on lipid accumulation in adipose tissue include, at least partially, reducing fatty acid transportation, fatty acid synthesis, and adipogenesis, as well as increasing the efflux of excess cholesterol. 

#### 2.2.4. Chitosan Modulates Adipokines Secretion 

Adipose tissue is a crucial endocrine organ and produces numerous adipokines, such as proinflammatory cytokines (TNF-α and IL-6), leptin, adiponectin, resistin and retinol binding protein 4 (RBP4), monocyte chemoattractant protein-1 (MCP-1/CCL2), epithelial neutrophil activating peptide (ENA-78/CXCL5), lipocalin 2 (LCN2), and secreted frizzled-related protein 5 (SFRP5), regulating multiple physiological functions, including energy metabolism and insulin sensitivity [76,77,78]. Dysregulation of those adipokines by obese and inflammatory adipose tissue may lead to decreased liver and muscle insulin sensitivity and, ultimately, systemic insulin resistance [79]. Liu et al. observed a reversal of the decrease in plasma adiponectin levels and increase in plasma leptin levels, along with an improvement of insulin resistance and impaired glucose tolerance, when high-MW chitosan (380 kDa) was administrated together with high-fructose-diet-fed rats [48]. COS may also increase plasma adiponectin levels, compared to high-fat diet-fed rats [80]. In STZ/nicotinamide-induced T2DM rats, the elevated TNF-α and IL-6, as well as lowered adiponectin, were also reversed by high-MW (830 kDa) chitosan treatment [47]. In obese diabetic *ob/ob* mice, COS supplementation improved BW gain, dyslipidemia, and hyperglycemia, as well as regulated a variety of adipokine expression, shown by an increase in adiponectin and a decrease in RBP4, resistin, TNF-α, and IL-6 [81]. Bai et al. reported that COS with a MW of 10 kDa suppressed the upregulated proinflammatory cytokines (TNF-α, IL-6, MCP-1), accompanied by a reversal of downregulated PPARγ expression in the liver, which might improve the impaired glucolipid metabolism in high-fat-diet-fed mice [82]. However, Neyrinck et al. demonstrated that diet-induced obese mice exhibited a reduction in circulating levels of IL-6, resistin and leptin, but not MCP-1, after supplementation of chitosan extracted from white mushrooms, and the lower resistin and leptin levels were related to lower fat mass development [83]. Altogether, the modulation of adipokine secretion by chitosan and its derivatives, including leptin, adiponectin, resistin, RBP4, MCP-1, TNF-α, and IL-6, may contribute to their benefits in lipid and glucose metabolism. Overall, these findings suggest that adipose tissue is involved in the chitosan-mediated glucose homeostasis by the modulation of secreting adipokines as well.

It is of note that some recently discovered adipokines are associated with obesity and insulin resistance. SFRP5, an anti-inflammatory adipokines, is an endogenous inhibitor of wingless-type family member 5A (WNT5A) signaling pathways and has been speculated to be beneficial to regulating obesity, insulin resistance, and T2DM [77,84]. Moreover, CXCL5, which belongs to the chemokine family mainly regulating the chemotaxis of inflammatory cells (i.e., neutrophils), can be stimulated by TNF-α and is believed to be a mediator in an inflammatory process that links obesity to insulin resistance in mice [78], although Yang et al. reported that the elevated CXCL5 levels were associated with hypercholesterolemia, but not insulin resistance, in Chinese subjects [85]. Nevertheless, LCN-2 has been shown to have diverse pathophysiological activities, and it can be induced by inflammatory, hyperglycemic, and obese status. Thus, LCN-2 has been suggested to be involved in the regulation of insulin sensitivity and glucose homeostasis [86]. In human subjects, elevated LCN-2 was positively correlated with obesity, insulin resistance, impaired glucose regulation, and T2DM [87,88]. Further studies are encouraged to clarify the roles of SFRP5, CXCL5, and LCN-2 in the anti-diabetic effects of chitosan. 

### 2.3. Antioxidative and Anti-Inflammatory Effects of Chitosan and Its Derivatives

Chitosan and its derivatives possess antioxidant properties; however, most studies revealed that high-MW chitosan had weak or no antioxidant activities [89]. On the contrary, COS with lower MW has strong free radical scavenging activity against superoxide and hydroxyl radicals, because their hydroxyl and amino groups can be easily activated after exposure, which helps the scavenging free radicals [90]. Menids et al. reported that COS decreased intracellular radicals and suppressed NF-κB gene promoter activity in vitro, which implied the potential of COS in regulating diseases related with oxidative and inflammatory stress [91]. In a high-fat diet mouse model, COS with a MW of 1.5 kDa increased the activity of superoxide dismutase (SOD), catalase, and glutathione peroxidase in the stomach, liver, and serum, indicating that the antioxidant defense system is triggered by COS administration to protect against oxidative stress induced by a high-fat diet [92]. Moreover, Yuan et al. showed that COS, which possesses antioxidant activities, including an increase in serum total antioxidant capacity and SOD activity and a decrease in malondialdehyde (MDA) content, can protect the pancreas from oxidative stress in STZ-induced diabetic rats, by improving the reduction of islet size, loss of β-cells, and nuclear pyknosis of β-cells [46]. They also observed that COS was capable of prohibiting the apoptosis of pancreatic NIT-1 β-cells in response to oxidative damage induced by STZ in vitro. Moreover, Katiyar et al. suggested that the antioxidative properties of COS are involved in the anti-diabetic effects of COS by regulating the antioxidant enzyme activities and reducing lipid peroxidation in alloxan-induced diabetic mice [27]. However, a study found that high-MW chitosan (740 kDa) caused a remarkable decrease in hepatic lipid peroxidation and a significant increase in SOD and GPX, whereas no such effects were observed after COS with MW 719 Da administration in high-fat-diet-induced hyperglycemic rats [93]. They explained that the antioxidant effects of high-MW chitosan might be through different mechanisms other than scavenging radicals, by activating hepatic antioxidant enzymes and then reducing the liver lipid peroxidation. Overall, these results indicate that the antioxidant activity of chitosan and its derivatives is one of the mechanisms underlying the benefit of chitosan and its derivatives on the treatment of diabetes, and holds great potential to be further studied.

Inflammation plays a critical role in the pathology of many chronic diseases, including diabetes [94]. Dietary supplementation of chitosan has been proven to show strong immunomodulatory and anti-inflammatory activities in vitro and in vivo [95,96]. Zhu et al. have demonstrated that COS performs its anti-inflammatory effect by suppressing the production of nitric oxide, IL-1β, and TNF-α in LPS-stimulated RAW264.7 macrophage cells via blocking NF-κB signaling pathway [97]. The anti-inflammatory effect of COS was also proven by the in vivo findings that COS ameliorated intestinal inflammation in dextran sulfate sodium-induced colitis via inhibiting NF-κB activation and suppressing TNF-α and IL-6 levels [98]. High-fat-diet-induced hepatic steatosis in mice was remarkably ameliorated by COS with MW 5000 Da administration via suppressing both inflammation and oxidative stress [99]. They showed that hepatic pro-inflammatory cytokines (TNF-α, IL-1β, and IL-6), neutrophils infiltration, and macrophage polarization were decreased by COS, along with the activation of AMPK and antioxidant enzyme gene expression [79]. Moreover, in the high-sucrose-induced hyperglycemic rats and high-fructose-induced T2DM-like rats, the plasma proinflammatory cytokines TNF-α and IL-6 levels were suppressed by high-MW chitosan administration [48,100]. These findings suggest that the anti-inflammatory effects of chitosan may play a role in its anti-diabetic potential as well. Nevertheless, proinflammatory cytokine IL-18 was reported to be elevated in newly diagnosed T2DM and prediabetic patients [101,102], and acute hyperglycemia was shown to induce the increase in plasma IL-6, TNF-α, and IL-18 concentrations [103]. Moreover, Zhang et al. reported that IL-18 signaling mediated pancreatic β cell development, proliferation, and insulin secretion [104]. Collectively, it suggests that IL-18 plays a role in glucose homeostasis, insulin secretion, and the development of diabetes. The effects of chitosan on the IL-18 expression in diabetes need to be further explored.

## 3. Application of Chitosan and Its Derivatives in Clinical Trials

Although several clinical trials studied the efficacy of chitosan, the benefits of chitosan and its derivatives on glycemic regulation are not conclusive, due to these clinical trials varying substantially regarding the design of clinical research, the dosage, and the duration of chitosan supplementation, and characteristics of the enrolled population. Notably, a pooled analysis with trial sequential analysis showed that supplementation with chitosan effectively reduced plasma TC and LDL-C levels, indicating that daily chitosan consumption might be a worthwhile dietary approach to prevent hypercholesterolemia [105]. A meta-analysis of randomized controlled trials further showed that body weight, body mass index (BMI) and body fat were significantly reduced by chitosan supplementation, particularly in obese and overweight individuals [106]. Moreover, Guo et al. conducted a meta-analysis showing that supplementation with chitosan at 1.6 ~ 3 g per day for at least 13 weeks can ameliorate the blood glucose level of diabetic and obese subjects, recommending the use of chitosan as a long-term dietary fiber supplement to improve glycemic control and enhance insulin sensitivity [28]. Hernández-González et al. reported that supplementation of chitosan for three months could enhance insulin sensitivity and reduce body weight, BMI, waist circumference, and TG in obese patients without dietary intervention [107]. Supplementation with COS for 12 weeks in prediabetic subjects has been found to improve the postprandial blood glucose levels, HbA1c, pro-inflammatory cytokines IL-6 and TNF-α, and plasma adiponectin [108]. Nevertheless, on should be cautious in interpreting the clinical results. Further large-scale and long-term clinical trials are needed to better describe the benefits of chitosan in the management of diabetes.

Some selected literature reports on in vitro, in vivo, and clinical trial studies for antidiabetic effects are summarized in Table 1.

## 4. Safety Evaluation of Chitosan and Its Derivatives

The U.S. Food and Drug Administration (U.S. FDA) has recognized that chitosan and COS are Generally Recognized as Safe (GRAS) and approved it as food additives [26]. Oral administration of chitosan in mice has indicated that the LD50 is more than 16 g/kg [35]. The safety of COS has been evaluated using toxicological methods in vitro and in vivo [109]. It was found that COS with an average MW of 1.86 kDa was non-mutagenic in the Ames test using *Salmonella typhimurium* mutant strains and non-genotoxic in the mouse micronucleus test and mouse sperm abnormality test. Furthermore, in an oral acute toxicity test, the maximum tolerated dose of COS was found to be more than 10 g/kg for both male and female mice; in a 30-day oral toxicity test in Sprague Dawley (SD) rats, the no-observed-adverse effect level (NOAEL) of COS was revealed to be equal to 3 g/kg body weight [109]. These findings suggested that short-term supplementation of COS could be safe when considering the possible levels of COS used in food systems or functional foods. Kim et al. also suggested that subacute oral toxicity of COS was low, and the NOAEL was considered to be over 2 g/kg in rats [110]. Liu et al. also demonstrated that normal SD rats supplemented with 5% COS for 12 weeks did not induce liver toxicity and abnormal lipid metabolism [111]. However, some concerns, such as the impurities or degradation products contained in chitosan and COS, may be a controversial issue for the application of chitosan and its derivatives in the dietary blend [29]. Several in vitro and in vivo studies have indicated that COS has varying degrees of side effects (summarized in Table 2) [112,113,114,115,116]. Ylitalo et al. indicated that mild and transitory nausea and constipation were observed in 2.6–5.4% of human subjects supplemented with dietary chitosan [117]. They further suggested that it would not be recommended to people allergic to crustaceans, although chitosan has been clinically well tolerated.

## 5. Future Prospect

Due to their unique physiochemical properties, broad safety profiles and diverse beneficial biological activities, chitosan and its derivatives are considered promising natural polymers that possess therapeutic potential in the management of diabetes. However, there are still several challenges and controversies that need to be addressed in the future. For instance, Yao et al. reported that supplementation with low-MW chitosan had no effect on glycemic control in STZ-induced diabetic rats [58]. Moreover, in GK spontaneously non-obese diabetic rats, glucose lowering effects were not observed after administration of COS, nor were lipid lowering effects [112]. It is generally accepted that a high-fiber diet is beneficial for managing metabolic syndromes, such as T2DM and/or hyperglycemia. The European Food Safety Authority (EFSA) Panel on Dietetic Products, Nutrition and Allergies (NDA) recommended an intake of 3 g of chitosan per day to achieve the beneficial physiological effect of maintaining normal blood cholesterol levels in adults [118]. Many hundreds of tons of dietary chitosan products in different forms, such as tablets, capsules, and dietary supplements, are consumed yearly in Europe and USA [119]. As a low-cost natural polymer with good biodegradability and biocompatibility, as well as having anti-diabetic properties, chitosan is a good candidate for further research in the prevention, treatment, and management of diabetes.

## 6. Conclusions

Chitosan and its derivatives have been shown to improve glucose and lipid homeostasis in diabetic rodents, which may provide a new option for diabetic intervention. A schematic diagram outlining the possible mechanisms involved in the anti-diabetic effects of chitosan and its derivatives is shown in Figure 1. Nevertheless, more scientific evidence is urgently needed from both animal studies and clinical trials to further understand its glucose regulating effects and safety.

## Figures and Tables

**Figure 1 marinedrugs-20-00784-f001:**
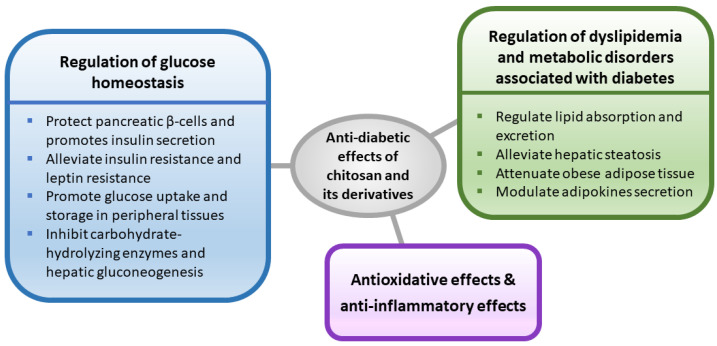
A schematic diagram outlining the possible mechanisms involved in the anti-diabetic effects of chitosan and its derivatives.

**Table 1 marinedrugs-20-00784-t001:** A summary of the selected literature on the antidiabetic effects of chitosan and its derivatives.

MW and DD of Chitosan/COS ^1^	Experimental Species	Routes of Administration	Doses and Durations	Antidiabetic Effects	References
Average MW 20 kDa ^2^	Obese diabetic KK-Ay mice	In drinking water	0.05%, 0.2% or 0.8% for 11 weeks	Anti-hyperglycemia; anti-hyperinsulinemia; anti-hypertriglyceridemia	Hayashi & Ito (2002) [35]
Average MW 1.5 kDa; DD = 86.5%	High-energy diet/streptozotocin-induced diabetic Sprague Dawley rats; in vitro β-cells	Oral (gavage)	500 and 1000 mg/kg for 8 weeks	Anti-hyperglycemia; improvement of glucose tolerance; protecting β-cells	Ju et al. (2010) [42]
Average MW < 1000 Da (GO2KA1) ^2^	*db/db* diabetic mice	In diet	4% for 42 days	Reduction in blood glucose and HbA1c; inhibition of carbohydrate hydrolysis enzymes	Kim et al. (2014) [43]
Average MW = 380 kDa; DD = 89.8%	High-fructose diet-fed Sprague Dawley rats	In diet	5% for 21 weeks	Improvement of impairment in glucose and lipid metabolism	Liu et al. (2015) [48]
Average MW < 1000 Da (GO2KA1) ^2^	In vitro (Intestinal Caco-2 cells and 3T3-L1 preadipocytes)	Culture medium	1 and 10 mg/mL	Inhibiting intestinal glucose digestion and transport; enhancing glucose uptake	Yu et al. (2017) [57]
Average MW < 1000 Da (GO2KA1) ^2^	Human subjects with prediabetes	Oral (capsule)	1500 mg per day for 12 weeks	Help control postprandial glucose	Kim et al. (2014) [108]

^1^ MW: molecular weight; DD: degree of deacetylation. ^2^ In these literature reports, information on the DD was not shown.

**Table 2 marinedrugs-20-00784-t002:** A summary of the selected literature on the adverse effects of chitosan oligosaccharide (COS).

MW and DD of COS ^1^	Experimental Species	Routes of Administration	Doses and Durations	Adverse Effects	References
Average MW 5000 Da; DD > 90%	Normal Wistar rats	In drinking water	0.5% for 6 weeks	Reduction in body weight and induction of liver and skeletal muscle mitochondrial toxicity	Teodoro et al. (2016) [112]
Average MW 5000 Da; DD > 90%	Goto–Kakizaki diabetic rats	In drinking water	0.5% for 6 weeks	Increase in blood aspartate transaminase (AST) and direct bilirubin	Teodoro et al. (2016) [112]
Average MW 719 Da; DD =100%	High-fat diet-induced obese Sprague Dawley rats	In diet	5% for 8 weeks	Increase in blood AST and alanine aminotransferase (ALT) and tumor necrosis factor-α (TNF-α)	Chiu et al. (2019) [113]
Low MW ^2^	Normal rabbits	Intravenous injection	7.1-8.6 mg/kg for 11 days	Decrease in appetite and increased serum lysozyme activity	Hirano et al. (1991) [114]
MW≤ 2000 Da; DD > 85%	Pregnant normal Wistar rats	Oral	50 and 150 mg/kg for 10 days	Teratogenic toxicity	Eisa et al. (2018) [115]
Average MW 3000 and 5000 Da; DD ≥ 90%	Human blood	In vitro	0.5 and 1 mg/mL	Hemolysis	Guo et al. (2018) [116]

^1^ MW: molecular weight; DD: degree of deacetylation. ^2^ In this study, a mixture of chitosan oligosaccharides with a degree of polymerization (DP) of 2–8 was used, but information on the MW and DD for COS was not given.

## Data Availability

Not applicable.

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
