# Peer review of "Antidiabetic Properties of Chitosan and Its Derivatives"

_marinedrugs, 2022, doi:10.3390/md20120784_

Round 1

Reviewer 1 Report

Thank you dear Editor for your invitation about reviews this paper title by

Antidiabetic Properties of Chitosan and its Derivatives.

So, This Review, is an adequately  subject your Journal.

but I have some comments.

English is required to be improved.

keywords : the first later should be capital letters.

The introduction was well-written, brief, and concise, and covered all the important keywords. The objectives of this study were also clearly stated and the highlights of the importance is very well understood. Well done, but the methodology needs more clearly. The aim of the research should also be better justified - in the introduction to the work.

- Please rewrite the introduction as some recent research related to the current topic must be included and then linked to the end of the introduction for the purpose of the current research to complete the cognitive and scientific communications

add some references like https://doi.org/10.3390/life12101579

 https://doi.org/10.3390/molecules27123744

https://www.mdpi.com/2311-7524/8/10/914

-The abbreviations must be defined or given full names at their first existence in the main

- finally, The manuscript needs thorough revision before it can be considered for publication.

Author Response

Reviewer #1

 English is required to be improved.

Response: Thank you for your valuable comments. We have revised and polished the English of the manuscript according to the suggestion of reviewer.

  1. keywords: the first later should be capital letters.

Response: Thank you for your comments. We have made the correction as follows.

Keywords: Chitosan; Chitosan oligosaccharide; Diabetes mellitus

  1. Well done, but the methodology needs more clearly. The aim of the research should also be better justified - in the introduction to the work.

Response: Thank you for your comments. The added sentence was included. The corresponding paragraph has been written as follows:

Over the past few decades, several emerging studies have been conducted to evaluate their benefits in diabetic control; however, the blood glucose-lowering effects of chitosan and its derivatives, as well as their mechanisms are still in progress to date. Diverse mechanisms might be involved in the anti-diabetic effects of chitosan, as such this review aims to discuss the latest research progress on their potential effects against diabetes from mainly in vivo rodent studies. Moreover, the possible mechanisms underlying these benefits of chitosan are particularly addressed. We also include some updated safety evaluation of chitosan and its derivatives. The information and data were collected by systematic survey of literature using PubMed, Web of Science, and Google Scholar, as well as by our research group. We hope this review will help the readers learn about the potential of chitosan and its derivatives on improving the glycemic control and maintaining the glucose homeostasis, which may be helpful for the future research in related fields, and then benefit the health of diabetic patients.

  1. Please rewrite the introduction as some recent research related to the current topic must be included and then linked to the end of the introduction for the purpose of the current research to complete the cognitive and scientific communications

Response: Thank you for your comments. We have added the information in this revised manuscript according to the suggestion of reviewer. The corresponding paragraph has been written as follows:

……Therefore, complementary and alternative therapies for DM management have been researched for years. A variety of bioactive products derived from fungi, plants, animals have been shown to possess anti-diabetic or hypoglycemic potential, such as plant-based curcumin, resveratrol, Teucrium polium aerial parts, and Artemisia campestris extract, as well as animal-based fish oil, milk casein, and chitosan [11-15]. Obesity, known for disordered lipid metabolism and insulin resistance, is recognized as one of the major risk factors for development of T2DM……

Over the past few decades, several emerging studies have been conducted to evaluate their benefits in diabetic control; however, the blood glucose-lowering effects of chitosan and its derivatives, as well as their mechanisms are still in progress to date. Diverse mechanisms might be involved in the anti-diabetic effects of chitosan, as such this review aims to discuss the latest research progress on their potential effects against diabetes from mainly in vivo rodent studies. Moreover, the possible mechanisms underlying these benefits of chitosan are particularly addressed. We also include some updated safety evaluation of chitosan and its derivatives. The information and data were collected by systematic survey of literature using PubMed, Web of Science, and Google Scholar, as well as by our research group. We hope this review will help the readers learn about the potential of chitosan and its derivatives on improving the glycemic control and maintaining the glucose homeostasis, which may be helpful for the future research in related fields, and then benefit the health of diabetic patients.

  1. The abbreviations must be defined or given full names at their first existence in the main

Response: Thank you for your comments. We have made the corrections in this revised manuscript according to the suggestion of reviewer.

  1. The manuscript needs thorough revision before it can be considered for publication.

Response: Thank you for your comments. We have carefully and thorough checked and revised the manuscript according to the suggestion of reviewer.

Reviewer 2 Report

Comments to the Authors of manuscript number: marinedrugs-2048730 entitled “Antidiabetic Properties of Chitosan and its Derivatives”.

It is a review about diabetes and chitosan as a biomaterial which can be used in its treatment.

 1. Introduction, part 1.1 presents shortly the problem relating to diabetes, but there is a lack its control by the two main parameters like glycohemoglobin or fructosamine. It should be added.

2. the part 1.2 and 2 should be connected into one part.

3. the part about chitosan derivatives should be separated and indicated

4. L 145 – as I wrote above, the role of glycohemoglobin in diagnosis of diabetes is not described, but here it is for the first time mentioned.

5. the part 3.1 and 3.2 describe different pathways by which chitosan can regulated glucose homeostasis and lipid economy, but not described resistin, adropin, lipocalin 2, IL 18, RBP4, CCL2, CXCL5

6. please add short information about SFRP5

7. the part of conclusion cannot have a form of discussion where the references are added. It should be rephrased.

Author Response

Reviewer #2

 Q1. ntroduction, part 1.1 presents shortly the problem relating to diabetes, but there is a lack its control by the two main parameters like glycohemoglobin or fructosamine. It should be added.

Response: We have added the information in the front section of 1.1. Diabetes Mellitus (DM) according to the suggestion of reviewer. The corresponding paragraph has been written as follows:

……It highlights the truth that DM has become an accelerating public health concern and a substantial burden on society [3]. In this regard, diagnosing prediabetes or diabetes early and accurately is therefore pivotal for physician to trigger the proper treatment and control the further progress. Fasting plasma glucose, 2-hour oral glucose tolerance test (OGTT) and glycated hemoglobin A1c (HbA1c) are the standard measures for diagnosis of T2DM. More recently, some nonclassical markers of hyperglycemia, including fructosamine, glycated albumin and 1,5-anhydroglucitol, are used as adjuncts to standard measures to obtain more detailed understanding of the alterations of glycemic control [4]. DM is characterized by hyperglycemia……

Q2 and Q3:

Q2. the part 1.2 and 2 should be connected into one part.

Q3. the part about chitosan derivatives should be separated and indicated

Response: Thank you for your comments and we answered Q2 and Q3 together. The paragraphs were rearranged following your suggestions. In detailed, we moved the content of section 2. Characterization of Chitosan to the section 1.2 Chitosan, and created a new section 1.3. Characterization of Chitosan Oligosaccharides (COS).

 Q4. L 145 – as I wrote above, the role of glycohemoglobin in diagnosis of diabetes is not described, but here it is for the first time mentioned.

Response: Thank you for your comments. We have corrected this issue as Q1 comment according to the suggestion of reviewer. Please see the response of Q1.

Q5 and Q6:

Q5. the part 3.1 and 3.2 describe different pathways by which chitosan can regulated glucose homeostasis and lipid economy, but not described resistin, adropin, lipocalin 2, IL 18, RBP4, CCL2, CXCL5

Q6. please add short information about SFRP5

Response: Thank you for your comments and we answered Q5 and Q6 together. The information was added in this revised manuscript according to the suggestion of reviewer. The corresponding paragraph has been written as follows:

2.2.2. Chitosan Alleviates Hepatic Steatosis

…….

It worthies noting that adropin, a peptide hormone secreted primarily by the brain and liver, is involved in glucose and lipid homeostasis [68]. It has been shown to be capable of protecting against NFALD and hyperinsulinemia in adropin-overexpressing transgen-ic mice fed with high-fat diet [69]. Gao et al. showed that adropin improved glucose tolerance and enhanced insulin sensitivity in diet-induced obese mice, along with the preferential utilization of carbohydrate for energy production in muscles [70]. Besides, they lately reported that hepatic glucose metabolism and hepatic insulin sensitivity were con-trolled by adropin [70]. Moreover, in a rat model of T2DM, adropin was found to re-duce blood glucose levels, improve insulin sensitivity, ameliorate lipid profile, and suppress inflammatory markers (TNF-α, IL-6 and iNOS) expression [72]. Thus, it would be interesting to see whether adropin plays a role in chitosan-mediated anti-diabetic effects.

2.2.4. Chitosan Modulates Adipokines Secretion

Adipose tissue is a crucial endocrine organ and produces numerous adipokines, such as proinflammatory cytokines (TNF-α and IL-6), leptin, adiponectin, resistin and retinol binding protein 4 (RBP4), monocyte chemoattractant protein-1 (MCP-1/CCL2), epithelial neutrophil activating peptide (ENA-78/CXCL5), lipocalin 2 (LCN2) and secreted frizzled‐related protein 5 (SFRP5), regulating multiple physiological functions, including energy metabolism and insulin sensitivity [76-78]. Dysregulation of those adipokines by obese and inflammatory adipose tissue may lead to decrease liver and muscle insulin sensitivity and ultimately systemic insulin resistance [79]. Liu et al. have observed a reversal of decrease in plasma adiponectin levels and increase in plasma leptin levels, along with improvement of insulin resistance and impaired glucose tolerance, when high-MW chitosan (380 kDa) are administrated together with high-fructose diet-fed rats [48]. COS may also have the ability to increase plasma adiponectin levels, compared to high-fat diet-fed rats [68]. In STZ/nicotinamide-induced T2DM rats, the elevated TNF-α and IL-6 as well as lowered adiponectin were also re-versed by high-MW (830 kDa) chitosan treatment [47]. In obese diabetic ob/ob mice, COS supplementation improved BW gain, dyslipidemia and hyperglycemia, as well as regulated a variety of adipokine expression shown by an increase of adiponectin and a decrease of RBP4, resistin, TNF-α and IL-6 [81]. Bai et al. reported that COS (10 kDa) suppressed the up-regulated proinflammatory cytokines (TNF-α, IL-6, MCP-1), accompanied by a reversal of down-regulated PPARγ expression in the liver, which may improve the impaired glucolipid metabolism in high-fat diet-fed mice [82]. However, Neyrinck et al demonstrated that diet-induced obese mice exhibited a reduction of circulating levels of IL-6, resistin and leptin, but not MCP-1, after supplementation of chitosan extracted from white mushrooms, and the lower resistin and leptin levels were related to lower fat mass development [83]. Altogether, modulation of adipokine secretion by chitosan and its derivatives, including leptin, adiponectin, resistin, RBP4, MCP-1, TNF-α and IL-6, may contribute to their benefits on lipid and glucose metabolism. Overall, these findings suggest that adipose tissue could be involved in chitosan-mediated glucose homeostasis by modulation of secreting adipokines as well.

It’s of note that some recently discovered adipokines are associated with obesity and insulin resistance. SFRP5, an anti-inflammatory adipokines, is an endogenous in-hibitor of wingless-type family member 5A (WNT5A) signaling pathways and has been speculated to be beneficial to regulating obesity, insulin resistance and T2DM [77,84]. Moreover, CXCL5, belongs to the chemokine family mainly regulating the chemotaxis of inflammatory cells (i.e. neutrophils), can be stimulated by TNF-α and is believed to be a mediator in inflammatory processes that links obesity to insulin resistance in mice [78], although Yang et al. reported elevated CXCL5 levels were associated with hypercholesterolemia but not insulin resistance in Chinese subjects [85]. Nevertheless, LCN-2 has been shown having diverse pathophysiological activities, and it can be in-duced by inflammatory, hyperglycemic and obese status. Thus, LCN-2 has been suggested to be involved in the regulation of insulin sensitivity and glucose homeostasis [86]. In human subjects, elevated LCN-2 was positively corelated with obesity, insulin resistance, impaired glucose regulation and T2DM [87,88]. Further studies are encouraged to clarify the role of SFRP5, CXCL5 and LCN-2 in the anti-diabetic effects of chitosan.  

2.3. Antioxidative and Anti-Inflammatory Effects of Chitosan and Its Derivatives

…… Nevertheless, proinflammatory cytokine IL-18 was reported to be elevated in newly-diagnosed T2DM and predi-abetic patients [101,102], and acute hyperglycemia was shown to induce an increase of plasma IL-6, TNF-α, and IL-18 concentrations [103]. Besides, Zhang et. al. reported that IL-18 signaling mediated pancreatic β cell development, proliferation, and insulin secretion [104]. Collectively, it suggests that IL-18 plays a role in glucose homeostasis, insulin secretion and the development of diabetes. The effects of chitosan on the IL-18 expression in diabetes need to be further explored.

Q7. the part of conclusion cannot have a form of discussion where the references are added. It should be rephrased.

Response: Thank you for your comments. We have made the changes according to the suggestion of reviewer. The corresponding paragraph has been written as follows:

  1. Future Prospect

Due to their unique physiochemical properties, broad safety profiles and diverse beneficial biological activities, chitosan and its derivatives are considered promising natural polymers that possess therapeutic potential in the management of diabetes. …… As a low-cost natural polymer with good biodegradability and biocompatibility, as well as having anti-diabetic prop-erties, chitosan is a good candidate for further research in the prevention, treatment and management of diabetes.

  1. Conclusion

Chitosan and its derivatives have been shown to improve glucose and lipid ho-meostasis in diabetic rodents, which may provide a new option for diabetic intervention……Nevertheless, more scientific evidences are urgently needed from both animal studies and clinical trials to further understanding of its glucose regulating effects and safety.

Round 2

Reviewer 2 Report

I do not have other comments.